# Primary Immune Thrombocytopenia and Essential Thrombocythemia: So Different and yet Somehow Similar—Cases Series and a Review of the Literature

**DOI:** 10.3390/ijms222010918

**Published:** 2021-10-09

**Authors:** Marta Sobas, Maria Podolak-Dawidziak, Krzysztof Lewandowski, Michał Bator, Tomasz Wróbel

**Affiliations:** 1Department of Hematology, Blood Neoplasms and Bone Marrow Transplantation, Wroclaw Medical University, Pasteura 4, 50-367 Wroclaw, Poland; maria.podolak-dawidziak@umed.wroc.pl (M.P.-D.); michal.bator@umed.wroc.pl (M.B.); tomasz.wrobel@umed.wroc.pl (T.W.); 2Hematology and Bone Marrow Transplantation Department, University of Medical Sciences, 60-569 Poznan, Poland; krzysztof.lewandowski@skpp.edu.pl

**Keywords:** immune thrombocytopenic purpura, essential thrombocythemia, platelet oscillation

## Abstract

This article collects several published cases in which immune thrombocytopenic purpura (ITP) is followed by essential thrombocythemia (ET) and vice versa. This surprising clinical condition is possible, but very rare and difficult to diagnose and manage. We have made an attempt to analyse the possible causes of the sequential appearance of ITP and ET taking into consideration the following: alteration of the thrombopoietin (TPO) receptor, the role of autoimmunity and inflammation, and cytokine modulation. A better understanding of these interactions may provide opportunities to determine predisposing factors and aid in finding new treatment modalities both for ITP and ET patients.

## 1. Introduction

In the past few years several cases of immune thrombocytopenia (ITP) which developed into essential thrombocythemia (ET), and in some instances, cases with ET followed by ITP have been reported [1,2,3,4,5,6,7,8].

ET together with polycythemia vera (PV) and primary myelofibrosis (PMF) belong to the Philadelphia-negative myeloproliferative neoplasms (MPN Ph-neg), which are characterized by overproduction of mature, largely functional cells originating from the transformed hematopoietic stem cell [9,10]. In simpler terms, ET is a persistent thrombocytosis for which no external causative factor can be determined. The etiology of ET is still not fully explained. In the majority of cases, so-called driver mutations in the gene *JAK2* (50–60%), *CALR* (20–35%) or *MPL* (5–10%) stimulate proliferation of megakaryocytes and overproduction of platelets (PLT) through constant activation of the JAK2-STAT3/5 pathway. However, there are still ET patients who do not present *JAK2*, *CALR* or *MPL* mutation; referred to as “triple negative (TN) patients”. Recently, new non-driver mutations (DNMT3A, TET2, IDH1/2, ASXL1, EZH2, IDH1/2, U2AF1, SF3B1, SRSF2, ZRSR2, TP53 among others) have been identified in ET patients. It is theorized that some may cooperate with the driver mutation, leading to the disease progression [9,10,11]. Nevertheless, neither of these groups of mutations are the origin of the disease but rather constituents of subsequent events. The initiating hit is still unknown [10,11]. One of the possible triggers for ET development could be chronic inflammation and/or autoimmune disorders including primary immune thrombocytopenia [10,12,13,14,15,16,17].

Immune thrombocytopenic purpura (ITP) is an acquired immune disorder of platelet destruction and its impaired production leading to isolated thrombocytopenia (platelet count <100.0 G/L) without another apparent cause. ITP previously stood for “idiopathic” or “immune” thrombocytopenic purpura, though the disease is no longer idiopathic and there are many patients without purpura, but the abbreviation “ITP” remains the same. There are two categories of ITP: primary, which does not have any other cause, and secondary, which occurs in association with lymphoproliferative disorders such as lymphoma or chronic lymphocytic leukaemia (CLL), immune dysregulation (systemic lupus erythematosus, HIV infection, HCV infection, and Helicobacter pylori). It seems that exposure of platelet antigens (due to infection, inflammation or molecular mimicry of viral antigens) and loss of tolerance (due to genetic disposition in immune related-genes, autoimmunity by comorbidities that implies a central dysfunction in tolerance, altered immune state) are required to induce ITP. In adults, primary ITP constitutes approximately 80% of the diagnosed patients, whereas the remaining 20% are affected by secondary ITP. The median age at diagnosis is about 55 years, and the annual incidence is approximately 3.3–3.9 per 100,000 adults and increases with age [18,19,20,21,22,23].

This review aims to analyse all the cases of ET followed by ITP or vice versa published so far and new ones under observation. We have also made an effort to explain the possible mechanisms of the sequential occurrence of two opposite platelet disorders. Although ET and ITP have different presentations (thrombocytosis versus thrombocytopenia), chronic inflammation and immune dysregulation appear in both [10,11,12,13,14,15,16,17,18,19,20,21,22,23]. There are some open questions related to this topic. What comes first: chronic inflammation or myeloid neoplasm? Can chronic inflammation trigger clinically relevant clonal haematopoiesis? How should we manage patients with this unusual clinical condition?

## 2. Platelet Maintenance

Regulation of platelet mass is important to avoid thrombocytopenia and thrombocytosis and their pathological consequences. The normal platelet count (PLT) is 150.0–400.0 G/L and the average platelet life span is 7 to 10 days. Platelets are produced by megakaryocytes in the bone marrow and are removed from circulation as they are activated and utilized at sites of vascular injury or cleared by reticuloendothelial macrophages predominantly in the spleen. Control and regulation of PLT number is complex. Thrombopoietin (TPO) is the key hormone responsible for platelet production acting via the TPO-TPO receptor system. Circulating platelets directly influence megakaryopoiesis and platelet production through Janus kinase 2 (JAK2) pathways. Currently there are two proposed mechanisms of TPO regulation. According to the older one called “the platelet sponging theory,” the TPO receptor (MPL) present on platelet surface binds free TPO, and internalizes it in order to degrade it. Therefore, when platelet counts are low, less TPO is removed, and more is available to stimulate megakaryopoiesis in the bone marrow (BM). Conversely, when the platelet count rises above a given set point, they act as a “sink” for TPO, binding and destroying it before TPO can stimulate megakaryopoiesis in the BM. The second mechanism implies that the Ashwell–Morell receptors (AMR, Ashwell–Morell receptor) on hepatocytes play an important role in the removal of PLTs from the bloodstream. This process is related to the changes of sialic acid content in proteins located in aged or senescent platelets. The sialic acid-reduced platelets interact with AMR on hepatocytes, activating the JAK-signal transducer and activator of transcription (STAT) signalling pathway to increase TPO transcription. AMR-mediated platelet clearance triggers hepatic TPO transcription and translation, and the new TPO is released [24,25]. Moreover, there are several loss of function (LOF) and gain of function (GOF) mutations in the TPO-MPL signalling axis described. Decreased expression or TPO-MPL (due to LOF mutation) produce thrombocytopenia while hyperactivation of MPL signalling (due to GOF mutation) results in pathological myeloproliferation [26,27].

## 3. Review of Cases Published in a Literature

We have identified in the literature five cases of ITP with subsequent development of ET (group 1, Table 1) [1,2,3,4,5]. ET was diagnosed several years after ITP (3.5, 4, 7, 13, and 21 years, respectively). *JAK2V617F* mutation was present in three and *CALR* mutation in one patient. As shown in Table 1, all patients were female with the range of age at time of diagnosis being between 14 and 72 years. With respect to the response to ITP therapy, four out of five were unresponsive to corticosteroids (prednisone or alternatively “pulses” of dexamethasone, DEX). Two patients with serious bleeding and very severe thrombocytopenia were given intravenous immune globulin (IVIg). None of these patients were treated with a thrombopoietin receptor agonist (TPO-RA). Three out of five ITP patients underwent splenectomy and had a good response.

Moreover, here we present a new case from our department, that has not yet been published. The patient was a 51-year-old female case of ITP diagnosed at the age of 27 (1997). She neither responded to treatment with prednisone, nor to azathioprine (AZT). The normalization of platelet count had been achieved after performing a splenectomy in 2004. In 2013, the diagnosis of systemic lupus erythematosus (SLE) was confirmed, and since then she has been treated with hydroxychloroquine at doses 200–400 mg daily. In May 2015 she developed venous thrombosis of the left forefinger, treated for 2 weeks with clopidogrel (75 mg/d) and sulodexidum. At that time, her platelet count increased up to 900 G/L, and the diagnosis of ET-TN was made on the basis of bone marrow histopathological examination and WHO criteria. She had no erythromelalgia during the entire follow-up period.

Nevertheless, the reverse scenario is also possible and interesting. Table 2 shows patients with ET with subsequent development of ITP [6,7,8]. Two out of three cases were female, with the range of age at time of diagnosis between 45 and 95 years. The response to the therapy was quite good, there was no need for splenectomy in any of the cases. Mutation *JAK2V617F* was positive in all three cases. The development of ITP in patients with ET is extremely rare. Moreover, in case No 1, anti-*Helicobacter pylori* antibodies were detected at the time of diagnosis of ITP, and it is equivocal whether ITP should be qualified as primary or secondary. However, the fast normalization of platelet count after IVIg indicates primary ITP. Case No 3 was even more exceptional due to the sequential occurrence of thrombotic thrombocytopenic purpura (TTP), ET and ITP.

A 42-year-old African American female was diagnosed in 1994 as having TTP, and successfully treated with plasmapheresis and vincristine. Her PLT remained in the normal range. However, in March 2001 her platelets count increased up to 1158.0 G/L and based on a bone marrow biopsy she was diagnosed with ET. Analysis of the *JAK2V617F* mutation additionally confirmed the diagnosis of ET few years later. She was treated firstly with aspirin (ASA) and Anagrelide (ANA), then due to ANA intolerance the drug was discontinued, and hydroxyurea (HU) was started. Her platelet count ranged from 270.0 (March 2001) to 726.0 G/L (October 2001). Suddenly, the PLT count dropped down to 12.0 G/L. HU was immediately withdrawn. Diagnosis of TTP was excluded and she was diagnosed as ITP and treated with DEX 40 mg for four days with no improvement in PLT count. On day four, IVIg was given, and her PLT numbers increased up to 290 G/L. ET occurred seven years after TTP, and ITP followed ET by 7 months and TTP by 7.5 years [8].

## 4. What Is New in Pathogenesis of ITP?

Chronic immune thrombocytopenia (ITP) is an autoimmune disorder by antiplatelet autoantibodies and/or by T cell-mediated platelet destruction, and impaired megakaryocyte function in the bone marrow [20,28,29]. For years, ITP has been explained by the presence of immunoglobulin G autoantibodies produced by B cells, but now it is considered predominantly as a T cell disorder. Antigen-specific T cells either destroy platelets in the spleen or impair platelet production in the bone marrow [23]. Patients with ITP produce anti-platelet IgG antibodies (and more rarely IgM or IgA antibodies), which bind to the platelets. Platelets which bind autoantibodies are subsequently recognized by phagocytes bearing Fcɣ- receptors inducing their impairment and/or degradation predominantly in the spleen. Autoantibodies most commonly target very abundant surface antigens such as glycoprotein (GP) αIIbβ3 (GPIIbIIIa) and GPIb-IX-V molecules, but autoantibodies against multiple platelet antigens are also commonly seen in ITP. However, in as many as 30% to 40% of the ITP patients, no detectible antibodies can be found [30]. As there is still no gold standard test for ITP diagnosis, more accurate diagnosis of ITP should be based on clinical and laboratory criteria, and include: platelet count <100 G/L, with the exclusion of other causes of thrombocytopenia; a low platelet count nadir <20 G/L; platelet count response to the therapy (corticosteroids, IVIg); a positive antiplatelet antibody test (if detected) [20].

Autoreactive antibodies are secreted by plasma cells and B cells. Moreover, CD19 + CD41hiCD38hi B-regulatory cells (Bregs), which promote peripheral tolerance, are also impaired in ITP. They fail to reduce CD4^+^ T cell activation and trigger the recruitment of CD4^+^CD25^+^Foxp3^+^ T regulatory cells (Tregs), a subtype of CD4^+^ T cells crucial for immune suppression and tolerance [31,32] via IL-10 secretion [33,34]. Enhanced cytotoxic T lymphocyte-mediated platelet destruction has been observed in ITP. The cellular immune response is also affected, leading to a decrease of Tregs and Bregs, which contributes to autoreactive plasma cell survival (supporting autoantibody production) and unbalanced ThCD4^+^ T cell subsets [35]. Cytotoxic CD8^+^ T cell are also activated, inducing platelet and megakaryocyte apoptosis as well as dysregulation of bone marrow niche homeostasis. Antiplatelet antibody production is under the control of T helper (Th) cells, and elevated antiplatelet T-cell reactivity has been found in ITP [36]. Th cell polarization observed in ITP has been attributed to increased Th1 and Th17 cells, decreased Th2 cells, and reduced or impaired Tregs [37]. Aberrant profiles have been correlated with the loss of immune tolerance in ITP patients. The pathogenic pattern is enhanced by the pro-inflammatory cytokine profile that consists of increased IFN-γ, IL-2 and IL-17 as well as decreased immunosuppressive IL-10, TGF-ß and IL-4, promoting antibody development [18,38,39]. Other inflammatory cytokines, including IL-6, IL-18, tumour necrosis factors α (TNF-α) and IL-7 could also be secreted by platelets and megakaryocytes [40,41,42].

## 5. Is ET Induced by Autoimmune Diseases?

According to Kristinsson et al., a prior history of autoimmune disease may significantly increase risk of MPN Ph-neg [43]. This study was performed on a total of 11,039 patients (6217 PV, 2838 ED, 1172 PMF and 812 MPN not otherwise specified) together with 43,500 population-based matched controls. Authors claim that in case of autoimmune disease there is a 20% increased risk of MPN development. It is worth noting that patients with a history of ITP, Crohn’s disease and polymyalgia rheumatica have a 2- to 3-fold increased risk for MPN development. However, the biggest risk is observed for patients with a history of giant cell arteritis (RR 5.9; 2.4–14.4), Reiter’s syndrome (RR 15.9; 1.8–142) and aplastic anaemia (RR 7.8; 3.7–16.7). Similar data were reported by Anderson et al. where analysis on Surveillance and Epidemiology and End Results (SEER) database was performed, 1017 MPN cases were analysed and a significant association between MPN development and localised scleroderma was observed [14]. In the case of patients who are diagnosed with MPN Ph-negative before autoimmune disease, it is possible that driver-mutations (*JAK2V617F*) stimulate the excessive production of cytokines, which may be responsible for chronic inflammation/autoimmune disease development [15].

Autoimmunity develops due to the breakdown of both central and peripheral tolerance against self-antigens. Genetic susceptibility (HLA genotype, cytokine polymorphisms, etc.) and environmental factors (e.g., infections, drugs, neoplasms) may be involved in autoimmunity induction [44]. Several mechanisms are implicated in the loss of immune tolerance, among an imbalance between the Tregs and effector T cells (Th1/Th17) [45,46,47]. Dysfunction of T cells is related, among others, to programmed cell death protein-1 (PD-1) overexpression [48,49]. The overexpression of PD-1 ligand 1, in *JAK2V617F*-positive cells (monocytes, myeloid-derived suppressor cells, megakaryocytes and platelets), is induced by the overactivation of JAK/STAT pathways [12,50,51]. Moreover, *JAKV617F*-positive cells are able to produce a large amount of reactive oxygen species (ROS), which affect T cells negatively [50,52]. Currently, JAK2 inhibitors are used in some autoimmune disease therapies (for example rheumatoid arthritis, psoriasis) [53]. Moreover, data in favour of the immunogenicity, especially *CALR* exon 9 and *JAK2V617F* mutation, provide justification for the development of potential immunotherapeutic targets [53,54,55,56,57,58]. A case of a severe autoimmune cytopaenia (Evans syndrome characterized by autoimmune haemolytic anaemia and immune thrombocytopenia) treated with JAK1/2 inhibitor (ruxolitinib) was also published. In this patient, Evans syndrome developed due to the gain of function mutation in the linker domain of Signal Transducer and Activator of Transcription (STAT)1, which provoked increased STAT1 phosphorylation, dysregulation of T helper 1 (TH1) and follicular T helper (TFH) development, and impaired TH17 response. One may be concerned with anaemia and thrombocytopenia related to therapy with ruxolitinib. However, in this case, therapy with ruxolitinib was able to control cytopaenia due to the reverse of immune dysregulation [59].

Other possible mechanisms of autoimmunity development could be related to autoantibodies directed at pro-inflammatory lysosphingolipid. These autoantibodies are present in 20% of MPN Ph-neg patients, including PMF and ET. These autoantibodies may be responsible for inflammation which overstimulate myelopoiesis via the JAK2/STAT5 pathway which facilitates the acquisition of JAK2 or CALR mutations and subsequent reactivation of the JAK2/STAT5 pathway [60]. There are also studies where an association between antiphospholipid antibodies (APA) and ET or PV diagnosis was observed. The first study was performed on 68 ET patients in 2002, with no data concerning *JAK2V671F* mutation status. ACA (anticardiolipin and anti-prothrombin antibodies) were found in 20 ET patients. The authors suggested that patients who form APA had a disordered immune surveillance system, which permits the emergence of a malignant disease [61,62].

Finally, a state of SOCS1 haploinsufficiency may be responsible for early onset of autoimmune diseases. Physiologically, the role of intracellular protein SOCS1 is to downregulate cytokine signalling by inhibiting the JAK-STAT pathway. In case of SOCS1 haploinsufficiency, lymphocyte hyperactivity due to increased STAT activation in response to interferon-γ, IL-2 and IL-4 occurs. This effect could be reverted by the JAK1/JAK2 inhibitor ruxolitinib [63].

## 6. Is ET Induced by Chronic Inflammation?

Patients with MPN Ph-neg present with a lot of “inflammation” symptoms (fatigue, pruritus, night sweats, weight loss, bone pain and/or fever, thrombotic complications). Moreover, in patients with ET, increased erythrocyte sedimentation rate (ESR), protein C reactive (CRP), serum ferritin level and neutrophil-to-lymphocyte ratio have been described [64,65,66]. Apart from CRP, interleukin (IL)-2, IL-6, IL-8, and soluble IL-2 receptor α (sIL-2Rα), tumour necrosis factor (TNF)-α, hepatocyte growth factor (HGF), platelet-derived growth factor, vascular endothelial growth factor and tumour growth factor β (TGF-β) are also overexpressed in MPN Ph-negative patients [12,16,17,60,67,68,69].

It was observed that “an inflammatory MPN phenotype” is induced by the stimulation of clonal MPN stem cells by some cytokines (IL6, IL8, TNF α and IFNα) in both animal models and in humans [17]. Release of proinflammatory cytokines, including the protein C reactive (CRP), has been related to clonal myeloproliferation [64,65] and the correlation between the JAK2V61F allele burden and CRP levels has been identified [69]. In that sense, an increase of WBC and platelet counts in MPN Ph-neg could be related to both clonal myeloproliferation and chronic inflammation [65,66].

It is worth adding that many of the inflammatory cytokines cause inferior survival of MPN patients [67,68]. Prolonged survival of cells such as fibroblasts and haematopoietic progenitors is related to activation of the JAK1/STAT3 or JAK2/STAT5 pathway by many of the proinflammatory cytokines [70,71].

Activation of the nuclear factor-κ-B (NF-κ-B) and JAK1/STAT among others are involucrate in enhancing inflammatory cytokine production [72,73,74]. Moreover, continuous release of pro-inflammatory cytokines stimulates the production of ROS, which in turn trigger the production of pro-inflammatory cytokines. Production of ROS could be related to the *JAK2V617F* mutation [12,50,52,75]. Higher levels of ROS have been described in patients with secondary acute myeloid leukaemia or myelofibrosis post-ET, which may suggest that ROS may be involved not only in ET development but could also be related to disease progression [64].

A potent inflammatory response in MPN patients is also induced by inflammasomes (the NLRP3 and AIM2). Inflammasomes are multiprotein complexes located in the cytosol. The main role of inflammasomes (as the part of the innate immune system) is to protect against invading pathogens but they are also implicated in the adaptive immune response via stimulation of the macrophages and regulation of Th17 cell differentiation [76]. More recently, they have been described in the cytokine storm in COVID-19 [77]. In the case of MPN patients, inflammasomes are implicated from cytokine storm to fibrosis [76].

According to the literature, development of several cancers, including hematologic ones, could be related to chronic inflammation [13,15,16,17,50]. There is some data that inflammation may explain clonal haematopoiesis of indeterminate potential (CHIP). CHIP is defined as a “presence of a clonally expanded hematopoietic stem cell caused by a leukaemogenic mutation in individuals without evidence of hematologic malignancy, dysplasia or cytopaenia”. CHIP is a potential premalignant phase of myelodysplastic syndrome (MDS), acute myeloid leukaemia or therapy-related myeloid neoplasms and according to the previous publications is associated with a 0.5–1.0% risk per year of leukaemia [78,79]. A cross-sectional study of 359 older adults showed that IL-6 and TNF serum levels were higher in people with CHIP than those without CHIP [80]. Mutations such as TET2 or DNMT3 are treated as CHIP-related mutants. It was observed in animal models that in MDS, a malignant clone with TET2 mutation, may proliferative or survive in a pro-inflammatory setting better than non-mutated haematopoietic stem cells (HSCs). Several autoimmune-inflammatory conditions (aplastic anaemia, ulcerative colitis, rheumatoid arthritis) have been reported to be enriched with CHIP [81]. Moreover, pro-inflammatory effects of CHIP are not limited to myeloid cells: T-cells may also have a role in perpetuating the inflammatory milieu. For this reason, inflammation seems to be a preclinically supported strategy to control CHIP development [78,82,83]. All these data need to be confirmed in humans.

The increased risk of cancer development could also be associated with the JAK2 46/1 haplotype of chromosome 9q [84]. The 46/1 haplotype is present in about 45% of the normal population and predisposes to additional mutations in JAK2 gene [13]. It was observed that the JAK2 46/1 haplotype is more frequent in patients with chronic inflammatory diseases and with some myeloid neoplasms including MPN [13]. Based on these data, there may be a connection between chronic inflammation and MPN development. According to the publication of Hermouet et al., the JAK2 46/1 haplotype should be treated as a marker of “inappropriate myelomonocytic response to cytokine stimulation leading to increased risk of inflammation, myeloid neoplasm, and impaired defence against infections” [84].

Nevertheless, there is still an open question about the chronology of inflammation and MPN [60]. Is excessive production of cytokines related to *JAK2/CALR/MPL/others* mutations or, rather, do *JAK2/CALR/MPL/others* precede inflammation? In a study performed by Allain-Maillet et al., where 40 cytokines and two cytokine receptors were analysed in MPN patients and in vitro (in human UT-7 cells genetically engineered to express MPN mutations), 26 cytokines were found to be overexpressed in MPN patients with *JAK2V617F* mutation and 23 were not linked to *JAK2V617F* expression [60]. In that sense, the inflammation process present in *JAK2V617F*-mutated MPN is mostly not related to *JAK2V617F* mutation: it may have preceded the acquisition of *JAK2V617F* mutation, and was reactive to the mutated clone or both.

## 7. Thrombocytopenia Due to ET

A recent prospective study found that about 8% of 435 patients presented an overt autoimmune disease (AID) at diagnosis of MPN [81]. It is a difficult challenge to explain how ITP “breaks” the neoplastic ET clone and starts to dominate. We can only look closely and carefully analyse three cases published so far. All of them were treated with hydroxyurea (HU) during the ET period, and as soon as the thrombocytopenia appeared, HU was stopped, but there was no platelet recovery. In case No 1 *Helicobacter pylori* infection could be eventually a trigger for ITP. The course of the disease is even more complex in case No 3 as TTP preceded the diagnosis of ET by 7 years and ITP developed seven months afterwards.

It is also important to note that all three patients were treated with HU, as cyclic thrombocytopenia has been described in a 66-year-old male with a 9-year history of PV associated with the *JAK2* mutation. About a year after HU was started, the platelet count dropped down to 15.0 G/L. The diagnosis of ITP had been considered, but treatment with corticosteroids, rituximab, and azathioprine was ineffective. Thrombocytopenia recurred with a period of about 28 days. It seemed that the patient was mistakenly diagnosed as having ITP. After discontinuation of HU, the platelet count rose up to 2907.0 G/L. Ruxolitinib was introduced, with dose escalation up to 20 mg twice daily. After 110 days on ruxolitinib, the platelet count ranged from 659 to 1211 G/L, and without cycling [85]. Other authors also reported large platelet count fluctuations in PV patients during treatment with HU [86,87]. Thus, reasons for sequential occurrence of ITP and ET are not yet clear, as the thrombocytopenia may occur by chance, or occur due to secondary related to infections (e.g., *Helicobacter pylori*), treatment (HU) or the presence of some concomitant disorders.

In all three cases, the mutation *JAK2V617F* was positive. The *JAK2V617F* mutation appears in the general population and may be present years prior to diagnosis of ET. Among 488 individuals from the Copenhagen General Population Study, 63 (0.1%) tested positive for the *JAK2V617F* mutation in the period 2003–2008. After 12 years 48 of 63 were studied for the second time, and at re-examination 20 patients had ET, 13 PV, seven PMF, and eight had no disease [88]. It was confirmed by the other group that *JAK2V617F* may be present for over a decade prior to diagnosis [89].

As we commented previously, the mutation *JAK2V617F* may constitutively activate inflammatory pathways in haematopoietic stem cells (HSCs) and progenitor cells. Moreover, *JAK2V617F* induces dysregulated phosphorylation of STAT3 and induces neutrophil activation. This promotes inflammatory pathways, including NF-κ-B and IL-6-GP130-JAK pathways [90]. NF-κ-B has a key role in inflammation and innate immunity as it also promotes tumour development [13]. Moreover, *JAK2V617F* mutation has an important role in generation of ROS—an inflammatory driver [50,52]. Interestingly, the results of this population-based study indicate that a loss-of-function polymorphism in the IL6 receptor reduces the risk of *JAK2V617F* mutation and MPN [91].

However, the *JAKV617F* mutation may exist not only in myeloid but also in lymphoid cells [92]. There are some cases of B-cell chronic lymphocytic leukaemia (B-CLL) with *JAK2V617F* mutation described: some of them presented first with MPN and afterwards progressed to CLL, others were diagnosed first with CLL and then with MPN. There were also two published cases of B-CLL with *JAK2V617F* mutation without any data of MPN [93]. Based on this, MPN and B-CLL may be originating from common progenitors. Nevertheless, the role of *JAK2V617F* mutation in the development of B-CLL still requires investigation. *JAK2V617F* mutation through constitutive activation of JAK-STAT signalling pathway in lymphocytes may lead to increased cell numbers and development of B-CLL. Moreover, *JAK2V617F* mutation allelic burden is probably a risk factor of developing a lymphoid malignancy (the risk was as high as 12-fold in cases of chronic lymphocytic leukaemia) in patients with MPNPh-neg [94].

As far as we know, patients with B-CLL have a 5–10% chance of developing autoimmune disorders, mainly cytopaenias. The most frequent cytopaenia is autoimmune haemolytic anaemia, followed by ITP. Pure red blood cell aplasia (PRCA) and autoimmune granulocytopenia are very rare. Autoimmune diseases can develop in patients with CLL due to loss of self-tolerance and aberrant T- and B-cell function, which results in autoantigen presentation by malignant CLL cells, antibody production by normal B cells and diminished immune surveillance through loss of regulatory T-cells [95,96]. Based on these data, when a patient has thrombocytopenia following ET, secondary thrombocytopenia must be excluded first.

## 8. Conclusions

Although ET and ITP have different origins, laboratory and clinical manifestation, they seem to have some features in common: the dysfunction of the TPO-MPL axis and chronic inflammation/immune dysregulation. Decreased expression or TPO-MPL function results in thrombocytopenia, whereas hyperactivation of MPL signalling (due to TPO receptor mutation or JAK2 mutation) results in myeloproliferative neoplasms (MPNs).

Only a few cases of sequential appearance of ET after ITP and even less of ITP after ET have been published so far. If some inflammatory and autoimmune disorders may predispose to ET to follow ITP, the vice versa clinical situation seems to be more complex, and the possible participation of drug induction as well as the prefibrotic stage of primary myelofibrosis in arising thrombocytopenia need to be considered. Possibly, modern laboratory techniques, including liquid biopsy and related biomarkers, could be used in differential diagnosis in cases in which there is an association of a MPN and another disorder [97]. Still, it is not fully understood how the pathophysiological mechanisms are involved in these platelet disorders, or how to predict the appearance of this clinical condition.

Currently suggested approaches to treating patients with ET and ITP are cytoreduction based on risk factors, straight control of cytopaenias, and withholding cytoreductive therapy with immediate initiation of immunosuppression when thrombocytopenia occurs. To the best of our knowledge, this is the first presentation of such a case series with a comprehensive review dedicated to this topic.

## Figures and Tables

**Table 1 ijms-22-10918-t001:** Essential thrombocythemia (ET) following primary immune thrombocytopenia (ITP). Bone marrow (BM) biopsy, dexamethasone (DXM) 40 mg/day × 4 days, intravenous immunoglobulin (IVIg), HD PDN: high dose prednisolone, ANA (anagrelide), HU (hydroxyurea), prednisone (PD), azathioprine (AZT), and acetylsalicylic acid (ASA). TTP (thrombotic thrombocytopenic purpura). (1): First line treatment, (2): second line treatment, (3): third line treatment, * allele burden, ** diagnosis of systemic lupus erythematosus (2013).

Authors	[1]	[2]	[3]	[4]	[5]	Observed Case
No	1	2	3	4	5	6
Gender	Female	Female	Female	Female	Female	Female
ITP diagnosis (in the age of)	January 2009 (14)	July 1992 (45)	March 2004 (72)	2004 (57)	2008 (64)	1997 (51)
ITP treatment	(1) DXM × 5 cycles (2) Splenectomy (August 2012): response	(1) PD × 3 (1st and 2nd relapse) After ET diagnosis: (2) 3rd ITP relapse: IVG + AZT (3) 4th ITP relapse: (4) PD + AZT: response	(1) PD: poor tolerance (2) PD +IVIG: poor efficacy and tolerance (3) splenectomy (March 2005): response	(1) HD PDN (2) splenectomy: response	(1) DXM: relapse DXM: response	(1) PD: no response (2) AZT: no response (3) Splenectomy (2004): response
ET diagnosis	August 2012	January 2013	January 2017	2011	February 2012	May 2015 **
Mutations	*JAK2V617F*: 11% *	*JAK2V617F*	*JAK2V617F*: 27% *	*CALR* exon 9 variant	*JAK2V617F*	TN
BM biopsy	Not done (patient refusal)	Compatible with ET (2013)	Compatible with ET (2017)	Compatible with ET	Compatible with ET	Compatible with ET
Interval between diagnosis of ITP and ET	3.5 years	21 years	13 years	7 years	4 years	18 years
ET treatment	No data	HU	HU	HU	HU + ASA	ANA

**Table 2 ijms-22-10918-t002:** Primary immune thrombocytopenia (ITP) following essential thrombocythemia (ET), Bone marrow (BM) biopsy, dexamethasone (DXM) 40 mg/day × 4 days, intravenous immunoglobulin (IVIG), ANA (anagrelide), HU (hydroxyurea), prednisone (PD), MPD: methylprednisolone, azathioprine (AZT), and acetylsalicylic acid (ASA).

Authors	[6]	[7]	[8]
No	1	2	3
Gender	Female	Male	Female
ET diagnosis	2014	2018	March 2001
Mutation	*JAK2V617*	*JAK2V617* (53% allele burden)	*JAK2V617*
BM biopsy	Compatible with ET	Compatible with ET	Compatible with ET
ET treatment	HU	(1). HU (2). ASA	(1). ASA (2). ANA—switching to HU due to intolerance
ITP diagnosis	70-year-old (2019)	95-year-old (April 2018)	42-year-old (October 2001)
Interval between diagnosis of ITP after ET	5 years (5 months after introduction of HU)	Few weeks	7 months
ITP treatment	(1): PD + IVIg	(1) MPD i.v. (2) PD	(1) DXM
Additional notes	Anti-*Helicobacter pylori* antibody test was positive		Diagnosis of TTP in 1994 (treated with plasmapheresis and vincristine)

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
