# Peer review of "Primary Immune Thrombocytopenia and Essential Thrombocythemia: So Different and yet Somehow Similar—Cases Series and a Review of the Literature"

_ijms, 2021, doi:10.3390/ijms222010918_

Round 1

Reviewer 1 Report

This review by Martas Sobas and colleagues is a well-written review of the literature exploring the unique relation between essential thrombocythemia (ET) and immune thrombocytopenic purpura (ITP).

As the authors write, the co-occurrence of these two phenotypes in one patient has seldom been reported, and may happen in any order, often years apart.

The presentation of the cases from the literature is clear, complete and the cases are relevant to the subject of the review.

The authors cover the topics of inflammation and auto-immunity in the context of MPNs and ITP in depth. However, from points 4 (line 148) to 8 (line 360), the authors often (but not always) cite other reviews to support their statements. It would be preferable to cite original works and not reviews (see below).

Some minor changes are also required in the introduction:

  • Lines 28-29 : “Simplifying, ET is a persistent thrombocytosis for which no cause can be determined.”
    “no cause can be determined” is misleading: the authors probably mean no “external causative factor can be determined” as the pathophysiology of ET is well known and often driver mutations (JAK2, CALR or MPL mutations) are found and known to be the cause of the disease. Only in the case of Triple Negative ET are no causes to the diseases found.
  • Line 33: CARL should be CALR
  • Lines 34-35 and Lines 36-37: the authors write about “new non-driver mutations” and “another recently described [mutation]” without ever defining them. It would be preferable to point out what are these mutations, even if the authors expand on their subject later in the manuscript.
  • Line 62: “… chronic inflammation and immune dysregulation appears” should be “…appear”.

As a conclusion, I would accept this manuscript for publication after minor revisions. The most important change to be made would be to cite original works than reviews themselves. When citing a review, the authors refer more to another one's opinion on the subject more than a scientific fact that would be reported in an original article. Doing so lowers the impact of the present work.

Author Response

Dear Reviewer

Thank you very much for all your sugestions. We changed citations according to your recommendations. 

Respect to the "minor changes"

  • Lines 28-29 : “Simplifying, ET is a persistent thrombocytosis for which no cause can be determined.” - it was changed
    “no cause can be determined” is misleading: the authors probably mean no “external causative factor can be determined” - it was changed
  • Line 33: CARL should be CALR - it was changed
  • Lines 34-35 and Lines 36-37: it would be preferable to point out what are these mutations, even if the authors expand on their subject later in the manuscript - it was changed
  • Line 62: “… chronic inflammation and immune dysregulation appears” should be “…appear”. - it was changed

Aditionally we performed english review of all manuscript. 

Reviewer 2 Report

The authors have submitted a well-written review + novel case report regarding the association of immune thrombocytopenia and essential thrombocythemia. The novelty of the paper is very high and I would recommend publication in the journal following the following corrections/improvement.

Minor points

line 79 where is degraded -> correct to where it is degraded.

line 84 plays -> correct to play

line 114 .the -> correct to . The (the requires capitalization)

line 115 correct Normalization to normalization

line 118 - venous thrombosis  - did she present erythromelalgia?

table 1 - add your case to this table too

line 140 turned on -> change to selected/prescribed

Major points

  • table 2 - referring to the case of Oda et al. (2019) - can the diagnosis of primary ITP be sustained if there was a proof of H. pylori infection? Wasn't it secondary ITP due to the aforementioned infection? Was the patient followed-up?
  • table 1 & 2 - please add the references in brackets so they will hyperlinked to the list of references and easier to access
  • Is ET Induced by Chronic Inflammation? - here you should also briefly discuss the role of the inflammasome and oxidative stress in ET. Additionaly, there are studies that have reported that ET subjects had high levels not only of CRP, but of ESR, serum ferritin and neutrophil-to-lymphocyte ratio. In addition, another trigger for blood cancers is oxidative stress and its crosstalk with low-grade chronic inflammation, as reactive oxygen species trigger the production of pro-inflammatory cytokines which in turn stimulate the production of reactive oxygen species, creating thus a vicious cycle. See: https://www.mdpi.com/1422-0067/22/2/561 https://doi.org/10.37358/RC.19.10.7581 https://doi.org/10.37358/RC.19.8.7435
  • Hopefully, in the near future, the diagnosis and management of MPNs, and in particular ET and PMF, especially because there is a need to clearly delineate between ET and prefibrotic PMF, might be based on the use of modern laboratory techniques, e.g., liquid biopsy and related biomarkers. Possibly, this could be a good technique in cases in which there is an association of a MPN and another disorder. See the following paper and possibly integrate it into your discussions: https://www.mdpi.com/2075-1729/11/7/677
  • Another important point of your paper is to discuss whether these patients were followed-up to check whether the diagnosis of primary ITP was sustained (versus secondary ITP). Check whether the authors of these case reports also excluded common causes of secondary ITP: lymphoproliferative disorders, infections (not only H. pylori but HIV, HCV, HBV), lupus/other autoimmune disorders, immune deficits (CVID), post-vaccination secondary ITP (influenza, MMR, hepatits A etc).
  • References - revise according to the style of the journal (American Chemical Society) - you can use an online formatting program (e.g. bibguru) for references if you don't have access to endnote/mendeley/zotero etc. 

Author Response

Dear Reviewer,

thank you very much for your very interesting comments. 

Respect to the corrections:

Minor points - it was corrected

Major points

  • table 2 - referring to the case of Oda et al. (2019) - anti-Helicobacter pylori antibodies were detected at the time of diagnosis of ITP, and it is equivocal whether ITP should be qualified as primary or secondary. But the fast normalization of platelet count after IVIg indicates primary ITP
  • table 1 & 2 - please add the references in brackets so they will hyperlinked to the list of references and easier to access - corrected
  • Is ET Induced by Chronic Inflammation? - here you should also briefly discuss the role of the inflammasome and oxidative stress in ET - performed. Additionaly, there are studies that have reported that ET subjects had high levels not only of CRP, but of ESR, serum ferritin and neutrophil-to-lymphocyte ratio - commented. In addition, another trigger for blood cancers is oxidative stress and its crosstalk with low-grade chronic inflammation, as reactive oxygen species trigger the production of pro-inflammatory cytokines which in turn stimulate the production of reactive oxygen species, creating thus a vicious cycle. See: https://www.mdpi.com/1422-0067/22/2/561 https://doi.org/10.37358/RC.19.10.7581 https://doi.org/10.37358/RC.19.8.7435 Thank you very much for your suggestions. Very interesting manuscript. Changes performed and the literature included. 
  • Hopefully, in the near future, the diagnosis and management of MPNs, and in particular ET and PMF, especially because there is a need to clearly delineate between ET and prefibrotic PMF (I fully agree with your comment which I included into the manuscritp). , might be based on the use of modern laboratory techniques, e.g., liquid biopsy and related biomarkers. Possibly, this could be a good technique in cases in which there is an association of a MPN and another disorder. See the following paper and possibly integrate it into your discussions: https://www.mdpi.com/2075-1729/11/7/677 - Thank you for your suggestion, very interesting manuscript, a short comment was included.
  • Another important point of your paper is to discuss whether these patients were followed-up to check whether the diagnosis of primary ITP was sustained (versus secondary ITP). Check whether the authors of these case reports also excluded common causes of secondary ITP: lymphoproliferative disorders, infections (not only H. pylori but HIV, HCV, HBV), lupus/other autoimmune disorders, immune deficits (CVID), post-vaccination secondary ITP (influenza, MMR, hepatits A etc). I am fully agree with you. Unfortunatelly, there are no additional information in published cases. 
  • References - revise according to the style of the journal (American Chemical Society) - you can use an online formatting program (e.g. bibguru) for references if you don't have access to endnote/mendeley/zotero etc.  - corrected

Round 2

Reviewer 2 Report

Well-done! The authors have accurately addressed my suggestions and the paper can be accepted for publication in its present form.